# Effects of the COVID-19 Pandemic on *Legionella* Water Management Program Performance across a United States Lodging Organization

**DOI:** 10.3390/ijerph20196885

**Published:** 2023-10-05

**Authors:** Jasen M. Kunz, Elizabeth Hannapel, Patrick Vander Kelen, Janie Hils, Edward Rickamer Hoover, Chris Edens

**Affiliations:** 1Division of Foodborne, Waterborne, and Environmental Diseases, National Center for Emerging and Zoonotic Infectious Diseases, Centers for Disease Control and Prevention, Mailstop H24-11, 1600 Clifton Road, Atlanta, GA 30333, USA; 2Division of Bacterial Diseases, National Center for Immunization and Respiratory Diseases, Centers for Disease Control and Prevention, Mailstop H24-6, 1600 Clifton Road, Atlanta, GA 30333, USA; qlr0@cdc.gov (E.H.); iek4@cdc.gov (C.E.); 3Division of Environmental Health Science and Practice, National Center for Environmental Health, Centers for Disease Control and Prevention, Mailstop S106-5, 4770 Buford Highway, Atlanta, GA 30341, USA; lup8@cdc.gov (P.V.K.); xmo2@cdc.gov (E.R.H.); 4Oak Ridge Institute for Science and Education, P.O. Box 117, Oak Ridge, TN 37830, USA

**Keywords:** *Legionella*, Legionnaires’ disease, water management, COVID-19

## Abstract

*Legionella*, the bacterium that causes Legionnaires’ disease, can grow and spread in building water systems and devices. The COVID-19 pandemic impacted building water systems through reductions in water usage. *Legionella* growth risk factors can be mitigated through control measures, such as flushing, to address stagnation, as part of a water management program (WMP). A national lodging organization (NLO) provided WMP data, including *Legionella* environmental testing results for periods before and during the pandemic. The statistical analysis revealed an increased risk of water samples testing positive for *Legionella* during the pandemic, with the greatest increase in risk observed at the building’s cold-water entry test point. Sample positivity did not vary by season, highlighting the importance of year-round *Legionella* control activities. The NLO’s flushing requirements may have prevented an increased risk of *Legionella* growth during the pandemic. However, additional control measures may be needed for some facilities that experience *Legionella* detections. This analysis provides needed evidence for the use of flushing to mitigate the impacts of building water stagnation, as well as the value of routine *Legionella* testing for WMP validation. Furthermore, this report reinforces the idea that WMPs remain the optimal tool to reduce the risk of *Legionella* growth and spread in building water systems.

## 1. Introduction

Legionnaires’ disease is a severe respiratory illness with pneumonia caused by the bacterium, *Legionella*. In addition to Legionnaires’ disease, there are two other types of legionellosis, or illness caused by *Legionella* bacteria. Pontiac fever is a milder respiratory illness that does not involve pneumonia and resolves without the need of medical intervention. Extrapulmonary legionellosis is rare and results from infection in areas outside of the respiratory system. *Legionella pneumophila* serogroup 1 or SG1, is more strongly associated with disease than the other serogroups and species [1]. However, many non-pneumophila species are known to cause disease. More than 95% of reported United States Legionnaires’ disease case patients are hospitalized, and the case fatality rate is 10%, which increases to about 25% for disease acquired in a healthcare setting [2,3]. Nearly 9000 cases of Legionnaires’ disease were reported in the United States in 2019, and the Centers for Disease Control and Prevention (CDC) estimates that there may be 2.3 times more cases than those reported, due to underdiagnosis [4,5]. Legionnaires’ disease follows a seasonal pattern, with incidence increasing during the warmer months [6]. The yearly healthcare costs are estimated at USD 402 million in the United States [5]. 

*Legionella* naturally occurs in water but can grow and spread in poorly maintained building water systems and devices. Water stagnation, inadequate disinfectant residuals, warm water temperatures, and the presence of sediment, scale, or biofilm are all factors that can increase the risk of *Legionella* growth and spread in a building water system. Exposure typically occurs through the inhalation of aerosolized water from plumbed devices containing *Legionella*. Hot tubs, cooling towers, and showerheads are examples of aerosol-producing sources known to be associated with disease [1,7,8]. Identified outbreaks are often linked to large or complex water systems, such as those found in hotels or resorts, hospitals, long-term care facilities, and cruise ships.

A water management program (WMP) identifies hazardous conditions and outlines the steps to reduce the risk of *Legionella* growth and spread and is recommended by organizations like the CDC and ASHRAE [9,10]. Developing and maintaining a water management program is a multi-step process that requires continuous review. Water management programs should be reviewed at least annually or when events occur, such as when data reviews indicate control measures are persistently outside of the control limits [9]. The COVID-19 pandemic had a dramatic impact on the built environment, including building water systems and devices, through reductions in water usage because of severely reduced occupancy in buildings [11]. During 2020, the U.S. lodging industry occupancy rates were historically low, at 44%, a 33.3% decrease from 2019 [12]. Building closures and changing occupancy in response to the pandemic also impacted water usage patterns at the public water distribution system level [13]. Reduced water usage can make systems unsafe for use and can create favorable conditions for *Legionella* growth, such as stagnation or depleted disinfectant residuals [14,15,16,17]. The recommended steps for returning to normal occupancy or reopening after a period of no or low water usage include using or creating a WMP, flushing, and ensuring all potable and non-potable devices are properly maintained, according to the manufacturers’ recommendations [16,18,19,20]. However, research gaps exist regarding the evaluation of flushing effectiveness and optimal flushing duration and frequency [11,20].

In this analysis, we evaluated the pandemic’s impact on building water quality and assessed flushing effectiveness in controlling *Legionella* growth as part of a WMP during long-term reductions in occupancy or water usage. We conducted this study through a partnership with a national lodging organization (NLO) with over 700 lodging facilities located in the United States, all of which were required by the NLO to have a WMP. Water management program data, including *Legionella* environmental testing results, were received for time periods both before and during the pandemic. This analysis provides needed evidence for the use of flushing to mitigate the impacts of building water stagnation, as well as the value of routine *Legionella* testing for WMP validation. Furthermore, this report reinforces the idea that WMPs remain the optimal tool to reduce the risk of *Legionella* growth and spread in building water systems.

## 2. Materials and Methods

The NLO provided CDC water management data from more than 700 managed properties based in the United States. The NLO provided access to temperature, disinfectant, and *Legionella* culture data collected as part of the facilities’ WMPs from 2018 to 2020. Of note, this NLO required a robust flushing protocol for all facilities as part of their respective WMPs during the pandemic. The flushing protocol required a minimum weekly flow of water in the main building water distribution system and bi-weekly flow in all individual point-of-use outlets in buildings that were closed or partially closed for 14 days or more. The NLO used a proprietary WMP software system to manage WMP program activities at individual facilities and capture associated data across the organization. The WMP data were collected and contained in three datasets from the NLO-managed properties, which covered 36 months (January 2018–December 2020) for temperature and chlorine data and 35 months (January 2018–November 2020) for *Legionella* culture data. The data contained results of *Legionella* culture testing performed by a third party at an Environmental *Legionella* Isolation Techniques Evaluation (ELITE) Program member laboratory using traditional spread-plate culture methods. NLO staff collected disinfectant and temperature data from each building’s hot- and cold-water premise plumbing systems. Prior to the CDC receiving the data, all facilities were deidentified. 

Routine environmental sampling for *Legionella* was required by the NLO at least annually at all properties to validate each facility’s WMP effectiveness. Environmental samples for *Legionella* testing were collected from either the building’s hot- or cold-water premise plumbing system. The test point for the cold water was the tap closest to the building’s incoming water main. The test point for the hot water was a tap most distal from the heater within the hot-water distribution system. The hot-water test point varied depending on occupied rooms. The sampling locations were determined by each facility according to the water distribution system design. The sampling locations within a facility varied over time due to accessibility. The culture results were reported as the detection of *Legionella pneumophila* serogroup 1 (SG1), *L. pneumophila* serogroups other than 1, other species of *Legionella*, or non-detection for any *Legionella*. The limit of detection was not available and was likely to have varied by sample according to collection and testing methods. *Legionella* detection was analyzed in favor of concentration, as qualitative results are more consistent across ELITE Program member laboratories than quantitative results [21]. For the purposes of this analysis, we combined *L. pneumophila* serogroups 2–15 and other species of *Legionella* into the non-SG1 category. The *Legionella* culture results were reported with the following two dates: login date (date sample was entered into a tracking system) and sample date (date the water sample was collected and marked on the sample label). The sample date was chosen for use in our data analysis due to a higher completion rate. If there was no sample date listed, then the login date was used. When the sample date and login date were different by 365 days or a multiple of 365 days, then there was a manual review for an assigned correction. The NLO policy set control limits for *Legionella* results according to quantity. If any *Legionella* was detected, it triggered a defined corrective action as part of the facility-specific WMP. Response activities were implemented in a tiered approach according to concentration values, with a detection of less than or 1 CFU/mL of *Legionella* triggering response activities. 

The hot- and cold-water disinfectant data included free chlorine and total chlorine measurements through categorical ranges. The data included both the free chlorine and total chlorine measurements, regardless of potable water disinfectant type for the facilities, which was not known by the NLO. The NLO policy set separate control limits for both free chlorine and total chlorine for hot and cold water. If a control limit was not met, it triggered a defined corrective action as part of the facility-specific WMP. The control limits for free chlorine in hot water and cold water were less than 0.4 mg/L and less than 0.5 mg/L, respectively. The control limit for total chlorine in both hot water and cold water was less than 1.0 mg/L. 

Facility water heater supply and hot-water return temperatures were also logged. The temperature of the water leaving the water heater (e.g., supply) had a lower control limit set by NLO policy at 124°F (51 °C) and an upper limit of 129°F (54 °C) or above for scalding-prevention purposes. In some instances, the upper limit may have been higher than 129°F (54 °C), in accordance with the facility-specific WMP. The temperature of the water in hot-water recirculation systems (when present) immediately prior to reheating (e.g., return) had a temperature control limit set by the NLO of no less than 118°F (48 °C). These data also identified if the facility had thermostatic mixing valves (TMV) installed. A TMV mixes hot and cold water in a central location (e.g., immediately following the water heater) or at point-of-use fixtures to prevent scalding. The dataset did not indicate TMV location. For our analysis, we excluded any reported temperature values above 165°F (74 °C), as those values were not consistent with potable water distribution systems in tourist accommodations and are likely to be erroneous.

The individual datasets (i.e., *Legionella* sampling, disinfectant, and temperature) remained separate and were categorized temporally by 3-month quarters before and during the pandemic. Quarters were defined according to seasonality of Legionnaires’ disease cases to reflect the ecology of *Legionella* rather than by annual cutoffs. Quarters were classified as Q1 (February 1–April 30), Q2 (May 1–July 31), Q3 (August 1–October 31), and Q4 (November 1–January 31). As a result, the earliest quarter, Q4 2017, does not include any data from 2017, but instead represents data from January 2018. The pandemic data period was defined as the first quarter (Q1) of 2020 to the last quarter (Q4) of 2020 (excluding January 2021). Quarters 2 and 3 represent the warmer months in this dataset, typically associated with an increase in Legionnaires’ disease. For SG1 and non-SG1 detection, the original dataset (n = 17729) was collapsed by the facility, year, quarter, pandemic period (before vs. during), *Legionella* season (5/1–10/31 vs. 11/1–4/30), water pipe (hot or cold), and SG1 detection. Observations were excluded from analysis if they were missing water line environmental sampling data. Therefore, 4399 (89.8%) SG1 and 4663 (90.1%) non-SG1 observations were included in the analysis. 

Risk ratios were calculated using a random-intercept generalized linear mixed model. This type of model allowed for each facility to have a unique intercept and multiple observations. Specifically, *Legionella* SG1 detection was modeled using a binomial distribution, while a Poisson distribution provided a better fit for *Legionella* non-SG1 detection. Data were aggregated across year and quarter to minimize the influence of oversampling by certain facilities. These models independently tested the relationships between *Legionella* detection (SG1 and non-SG1) and pandemic period (before vs. during), water pipe (hot or cold), and “*Legionella*” season (5/1–10/31 vs. 11/1–4/30). These explanatory variables were selected according to *Legionella* ecology and available data. The main effects, significant at ≤0.05, were deemed to be of interest and have been discussed within the text. Multiple comparisons comprising the interaction term(s) were adjusted using the Scheffe test to minimize the Type I error rate. These analyses were conducted using SAS 9.4 (SAS Institute, Cary, NC, USA, https://www.sas.com).

## 3. Results

### 3.1. Environmental Sampling for Legionella

The NLO data contain 17,729 environmental samples from the 725 facilities tested for *Legionella* from Q4 2017 to Q4 2020, with *Legionella* detected in 1608 (9%) samples. Of the samples that tested positive for *Legionella*, 411 (26%) tested positive for SG1, 1252 (78%) tested positive for non-SG1, and 55 (3%) tested positive for both SG1 and non-SG1. Differences were observed when considering the overall results for hot and cold water. The NLO collected 10,598 hot-water environmental samples from 722 facilities and 5656 cold-water environmental samples from 715 facilities. *Legionella* growth is typically associated with hot-water systems, as the temperatures more commonly fall within the favorable range for *Legionella* growth (77–113 °F, 25–45 °C) (Table 1). There was some variation in the number of environmental samples tested by each facility (Table 2). For example, 11 facilities accounted for 46% (n = 190) of all of the SG1-positive samples and 2 of those 11 facilities accounted for 21% (n = 88). The positivity rate did not correspond with the sampling rate (Figure 1). See Appendix A for additional figures depicting the number of facilities by the number of environmental samples taken and *Legionella* percent positivity.

### 3.2. Pandemic Association with Environmental Sampling for Legionella and Positivity 

The COVID-19 pandemic (Q1 2020 to Q4 2020) impacted water management activities, including the total number of samples collected. The facilities collected 6421 samples during the pandemic year, compared to 5465 and 5765 samples in 2018 and 2019, respectively. As anticipated, significantly fewer facilities (Z-score = −1.64, one-tail *p* = 0.043), 149 of 696 (21%), reported at least 1 *Legionella*-positive environmental sample before the pandemic, compared to 175 of 692 (25%) facilities during the pandemic. Ninety-four facilities (13%) had at least one positive sample both before and during the pandemic. The sample percent positivity of the 94 facilities was 28% for all of the samples taken both before and during the pandemic, 22% for the samples taken before the pandemic, and 37% for the samples taken during the pandemic (McNemar’s Test, *p* < 0.001). These positivity values are significantly higher compared to the facilities with *Legionella* detected only before the pandemic (10%, Chi-square = 179.3, *p* < 0.001) and to facilities with *Legionella* detected only during the pandemic (15%, Chi-square = 127.3, *p* < 0.001) (see Appendix A). There was an increase in facilities in Q2 and Q3 of 2020 that reported at least 1 *Legionella*-positive environmental sample, 170 of 689 (25%), compared to facilities in Q2 and Q3 of 2018 and Q2 and Q3 of 2019, which were 82 of 603 (14%) and 94 of 639 (15%), respectively. A total of 86 of the 170 facilities (51%) with at least 1 *Legionella*-positive environmental sample in 2020 also reported at least 1 positive sample in either Q2 and Q3 of 2018 or 2019. 

The majority (56%) of the *Legionella*-positive environmental samples were reported during the pandemic, whereas only 36% of all of the samples were collected during the pandemic. Specifically, 38% of the positive environmental samples were reported in Q2 of 2020, whereas only 28% of all of the environmental samples were collected in the same quarter. The percentage of positive environmental samples seen in the 2nd and 3rd quarter of 2020 increased by 219% for SG1 and 184% for non-SG1 from what was observed in 2019. This occurred when there was only a 20% increase in sampling. 

The likelihood of an environmental sample testing positive for SG1 during the pandemic was 2.06 times greater than before the pandemic (95% CI (1.47, 2.89)) (Table 3). In comparison, the likelihood of an environmental sample testing positive for non-SG1 during the pandemic was 2.12 times greater than before the pandemic (95% CI (1.74, 2.59)) (Table 4). Additionally, the cold-water environmental samples were just as likely as the hot-water samples to test positive for SG1 or non-SG1 during the pandemic (Table 3 and Table 4). As a result, cold water was primarily responsible for the increased risk of environmental samples testing positive for *Legionella* during the pandemic. However, overall, a hot-water sample was more than 1.5 times as likely to test positive for SG1 or non-SG1 than a cold-water sample (Table 3 and Table 4). While we observed a modest increase in the number of facilities that tested positive during the pandemic, among the facilities with any positive results, the number and percentage of positive environmental samples increased. A detailed account of the environmental sample positivity by pandemic period and water system is included in Appendix A.

### 3.3. Facilities with Persistent Legionella Detections 

Some of the facilities saw *Legionella*-positive environmental samples in multiple quarters. A total of 123 (19%) facilities had *Legionella*-positive environmental samples in more than 1 quarter. Most of the facilities with positive results for *Legionella* in multiple quarters, 97 (79%), had samples that were positive for non-SG1 in more than 1 quarter, while 33 (27%) facilities had samples that were positive for SG1 in more than 1 quarter. Nearly half, 27 of 65 (42%), of the facilities with SG1-positive hot-water samples had positive samples appear in multiple quarters. Of the 178 facilities that tested positive for non-SG1 in their hot water, 78 (44%) had environmental samples that were positive for non-SG1 in multiple quarters. A total of 3 (7%) of the 45 facilities that detected SG1 in their cold water had positive environmental samples in multiple quarters. Of the 122 facilities that detected non-SG1 in their cold water, 39 (32%) had environmental samples test positive for non-SG1 in multiple quarters. 

### 3.4. Seasonal Environmental Sample Legionella Positivity Patterns 

Though the sampling rate was different throughout the year, the positivity remained comparable. There was no evidence that the sample positivity risk was related to the seasonality of when the sample was collected (Table 3 and Table 4). When the environmental samples were divided between the cooler months (Q1 and Q4) and warmer months (Q2 and Q3), the positivity was 10% and 9%, respectively. The total number of samples tested in the warmer months (n = 16,021) was 9.4 times higher than that of the cooler months (n = 1708). Most of the facilities that had *Legionella*-positive environmental samples in the cooler months, 33 of 40 (83%), also had persistent detections across multiple quarters. This is in comparison to the facilities that tested positive for *Legionella* in the warmer months, 122 of 222 (55%), that also had persistent detections across multiple quarters. 

### 3.5. Chlorine Levels in the Facilities’ Water Systems and Temperature of Facilities’ Hot-Water Systems

There was little difference observed between the percentage of hot- and cold-water samples where free and total chlorine were within their respective control limits before and during the pandemic (Table 5). Similarly, the percentage of supply and return temperature readings that fell below the control limit before and during the pandemic did not differ at 4% (Table 6), likely due to the facilities’ ability to monitor and control the water heater supply and return temperatures, as part of their WMP.

## 4. Discussion

### 4.1. Pandemic Impacts on Sample Positivity and Flushing Effectiveness 

Routine flushing is a recommended practice as part of a WMP to maintain water quality parameters within the control limits during periods of no or low water usage. However, a lack of real-world evidence exists on the use of flushing as an effective control measure to reduce the risk of *Legionella* growth during long-term periods of unusually low water usage [11,20]. Long-term events can include seasonal shutdowns commonly experienced in lodging or educational settings. During the COVID-19 pandemic, the NLO implemented an organization-wide flushing policy in response to decreased occupancy and widespread shutdowns. As a result, the data collected as part of the facilities’ WMP activities enable the observation of the potential impact of flushing as a control measure to reduce the risk of *Legionella* growth and spread during periods of low or no water usage. Flushing is featured in numerous guidance documents developed to manage building water systems in response to the pandemic’s impact on building occupancy and water usage [16,18,19,20]. The experts agree that flushing is a key control measure to manage water system stagnation; however, a scientific consensus has not yet been established regarding specific elements, such as flushing duration and frequency [20,22]. The NLO responded to the drastic occupancy reductions across the organization by requiring the implementation of robust flushing protocols in April of 2020, as part of facilities’ WMPs. The NLO flushing procedure ensures a minimum weekly flow of water in the main building water distribution system and bi-weekly flow in all of the individual point-of-use outlets. The procedure at the time of implementation applied to the buildings that were closed or partially closed for 14 days or more. The facility staff flushed the guest room fixtures, including the bathroom sink, tub spout, bidets, showerhead, handheld shower wand, and bar or kitchen sink for 5 min. The toilets were also flushed twice. In addition, if the facility was closed for more than 7 days, the staff flushed all of the drinking fountains, public restrooms, water feeds to ice machines, kitchens, back-of-house sinks, and laundries. The guest rooms on the odd-numbered floors were flushed on the first and third weeks of the month and the guest rooms on the even-numbered floors on the second and fourth weeks. 

An increased routine of environmental sampling for *Legionella* and continuous facility WMP performance reviews by the NLO played an important role in determining if the flushing procedures were meeting the WMP goals. The flushing protocol overall was successful in preventing an increase in the risk of *Legionella* growth in most of the facilities. Some of the facilities with before-pandemic *Legionella* detections showed an increase in the percentage of environmental samples testing positive for *Legionella*, despite robust flushing. We observed a greater percent positivity for the samples collected from the facilities with *Legionella* detections both before and during the pandemic, relative to the facilities with positive results only before or during the pandemic. Furthermore, among those facilities with *Legionella* detections both before and during the pandemic, the percent positivity increased from before to during the pandemic, indicating that these facilities with previous positive results were responsible for the observed overall increase in the percent positivity. These observations indicate that flushing, as part of a comprehensive WMP, may not adequately control *Legionella* growth during periods of low or no water usage in those facilities with previous *Legionella* detections. Additional control measures, including engineering interventions, may be necessary in those facilities for which increased flushing did not adequately control the *Legionella* growth. 

The changes in building occupancy and water usage altered the demand on public water systems during the COVID-19 pandemic, potentially impacting the risk for *Legionella* growth in areas with reduced water flow rates or disinfectant residual loss [13,23]. All of the NLO facilities relied on consistent disinfectant levels delivered by the municipal water provider as a key WMP control measure in the hot- and cold-water systems. The cold-water systems without supplemental disinfection on-site were vulnerable to water containing *Legionella*, due to the difficulty in applying additional control measures. Overall, the NLO cold-water was approximately three times more likely to test positive for SG1 or non-SG1 during the pandemic than before the pandemic, and was just as likely to test positive for SG1 or non-SG1 as the hot-water system during the pandemic. These findings suggest that the building cold water was the primary driver of the increased risk of samples testing positive for *Legionella* during the pandemic. Furthermore, this underscores the importance of monitoring the incoming water quality by setting a cold-water control point that includes water quality parameter monitoring and consideration of *Legionella* testing. 

### 4.2. Water Management Program Considerations

Analyzing and sharing WMP data at an organizational level can improve policies driving water management practices and reduce the risk of *Legionella* growth and spread in facilities. The systematic collection and analysis of water quality parameter data is an essential element of WMP success, as it can further validate the effectiveness of control measures and allow for continuous program improvement [24]. The methods used and lessons learned in this NLO analysis can improve the implementation and refinement of WMPs. The size and richness of the dataset afforded us the opportunity to make important observations about the NLO’s policies for facility-specific WMPs that can benefit public health and water management stakeholders. Managing an organization-wide policy for facility WMPs, including routine environmental *Legionella* testing for WMP validation, presents unique challenges, due to the number, size, and complexity of numerous building water systems. To this point, the authors have identified both variability in the number of validation samples tested at facilities and overrepresentation in the positive environmental samples of the facilities with persistent *Legionella* detections. 

The NLO’s environmental sampling for the *Legionella* rate reflects the seasonal pattern of Legionnaires’ disease, with most of the sampling occurring during the warmer months. This represents a prudent WMP practice in accordance with the seasonal increase in LD cases. However, these data reveal that the facilities detect *Legionella* across all quarters, and environmental sample positivity did not significantly vary between the warmer and cooler months. The lack of evidence that sample positivity risk is related to seasonality suggests that the focus on mitigating *Legionella* growth risk should not decrease in the cooler months.

Persistent *Legionella* detections across sampling events, or the presence of multiple *Legionella* species or serogroups, may indicate that the existing WMP control measures are inadequate, and may be more associated with Legionnaires’ disease. Even in a well-controlled system, low levels of *Legionella* can be found in a few sampling locations [20,23,25]. Furthermore, setting testing frequency protocols that are responsive to environmental sample results, particularly in the case of changes to the incoming water quality, will increase comparability, result in resource savings, and may lead to the identification of facility *Legionella* issues. For example, it may be advantageous to increase the sampling quantity and frequency following *Legionella*-positive results and the implementation of corrective actions. Conversely, the facilities for which the water quality parameters, including *Legionella* test results, are consistently within the control limits may benefit from sampling at a lower frequency.

The findings in this study are subject to important limitations. The NLO did not have water quality data collected from outside of their facilities. To minimize this confounder, the cold-water test point sample served as a proxy for source water and the location was chosen to be as proximal to the building incoming water main as possible. The cold-water test point was potentially subject to building-specific *Legionella* growth risk factors, such as reduced water usage, which was commonly experienced during the pandemic. The limits of the detection for *Legionella* results were not available and likely varied by sample. Although the limit of detection was unknown, all of the samples were tested for *Legionella* using traditional spread-plate culture methods by ELITE Program member laboratories. The collection of the temperature and disinfectant residual data was subject to human error. Some information was missing or improbable (e.g., a cold-water temperature of 200 °F, 93 °C). The disinfectant residual measurements were documented in categories instead of entering the data as discrete values. The non-detected values for disinfectant residuals were assigned to a category that also included low but detectable values. In addition, the NLO facilities did not list the disinfectant type used by the municipal water utility. Thus, the free and total chlorine values may not have been applicable, given the type of disinfectant (e.g., chlorine, monochloramine, or chlorine dioxide) used by the water utility. These limitations did not allow for the type of analysis to determine whether the disinfectant levels were noticeably different during the pandemic (low or no water use) than before the pandemic when the water use was normal. It is not clear if the observed lack of association is due to a lack of change in disinfectant residual measures from before to during the COVID-19 pandemic or if it is because the changes did occur but were not detected with the inclusion of the non-applicable values. 

Additional geographic considerations in incoming water quality, temperature, and facility-specific occupancy levels could not be made, as the location of the individual facilities was unknown. Additional research around *Legionella*, disinfectant residuals, and temperature in real-world conditions should be explored. More research is needed on the effectiveness of flushing and optimal flushing techniques, as well as the implementation of additional control measures for instances where flushing alone cannot prevent *Legionella* growth and spread within a building water system.

## 5. Conclusions

The COVID-19 pandemic impacted building water systems and devices through reductions in water usage, due to building closures or reductions in occupancy. WMP data from an NLO enabled the analysis of water quality parameter measurements before and during the COVID-19 pandemic. The risk of an environmental sample testing positive for *Legionella* during the pandemic doubled compared to the before-pandemic risk. Increased *Legionella* positivity was observed at the building’s cold-water entry test point in some of the NLO tourist accommodation facilities in conjunction with the COVID-19 pandemic. This increased risk occurred despite the robust flushing protocols implemented in accordance with facility-specific WMPs. The increased sample positivity was driven primarily by the facilities that had *Legionella*-positive environmental sample results prior to the COVID-19 pandemic. Overall, there was only a small increase (21 to 25%) in the number of facilities detecting any *Legionella* before and during the pandemic. This suggests that robust flushing programs may be adequate for many facilities during periods of decreased occupancy; however, other facilities may require additional control measures and possible engineering-related interventions to prevent *Legionella* growth and spread. The chlorine and temperature levels were stable, though we acknowledge the limitations of the disinfectant data. In our analysis, *Legionella* culture testing provided valuable information about a building’s cold-water entry quality and WMP performance that was not necessarily apparent through temperature and disinfectant residual measurements.

This analysis provides needed evidence for the use of flushing to mitigate the impacts of building water stagnation, as well as the value of routine *Legionella* testing for WMP validation. Furthermore, this report reinforces the idea that WMPs remain the optimal tool to reduce the risk of Legionella growth and spread in building water systems. The data collected through water management activities and validation procedures, including *Legionella* test results, provide critical information and enable performance improvement. These data were particularly valuable during the COVID-19 pandemic when tourist accommodation occupancy levels decreased dramatically for prolonged periods. These data, shared with public health partners by an NLO, can inform WMP optimization industry wide and contribute to our understanding of effective *Legionella* control measures.

## Figures and Tables

**Figure 1 ijerph-20-06885-f001:**
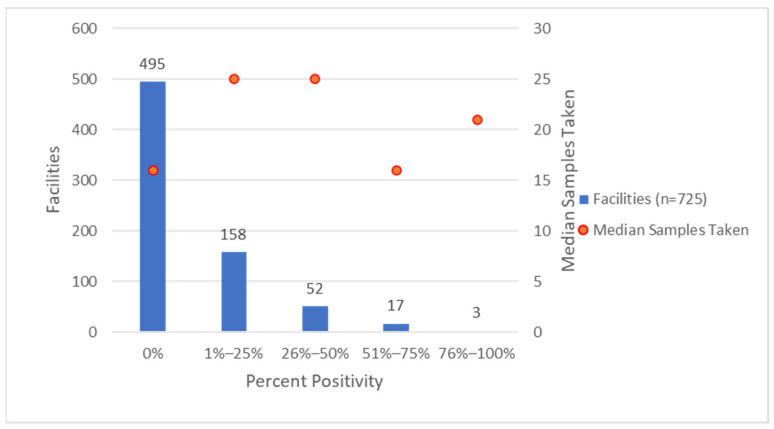
Number of Facilities and Median of Total Environmental Samples Tested by Facilities in *Legionella* Percent Positivity Category.

**Table 1 ijerph-20-06885-t001:** Number of Facilities with *Legionella* Detections.

Facilities with	Hot Water (n = 722)	Cold Water (n = 715)	Both (n = 725)
Any SG1 or non-SG1 detections ^1^	201 (28%)	143 (20%)	230 (32%)
Any SG1 detections	65 (9%)	45 (6%)	85 (12%)
Any non-SG1 detections	178 (25%)	122 (17%)	209 (29%)
Any SG1 and non-SG1 detections	42 (6%)	24 (3%)	64 (9%)

^1^ SG1 is *Legionella pneumophila* serogroup 1 and non-SG1 is *Legionella pneumophila* serogroups 2–15 and other species.

**Table 2 ijerph-20-06885-t002:** Descriptive Statistics of the Environmental Samples Tested for *Legionella* for 725 facilities.

Average Number of Samples Tested by Facility	24.45	Median Number of Samples Tested by Facility	18
Standard Error	0.79	Range of samples tested by facility	3–254
Standard Deviation	21.38	Sum of samples tested	17729

**Table 3 ijerph-20-06885-t003:** Relative Risk(s) of Environmental Samples Testing Positive for Legionella (SG1).

Effects	Risk Ratio (95% CI)	*p*-Value	Risk Ratio (95% adj.^1^ CI)	adj.^1^ *p*-Value
COVID (*p* < 0.001)				
Before COVID (3 January 2018–30 January 2020)	ref	--		
During COVID (1 February 2020–17 November 2020)	2.06 (1.47, 2.89)	<0.001		
Water pipe (*p* < 0.001)				
Hot-water pipe	1.93 (1.38, 2.68)	<0.001		
Cold-water pipe	ref	--		
Season (*p* = 0.481)				
Cold seasons (11/1–4/30)	1.21 (0.71, 2.05)	0.481		
Warm seasons (5/1–10/31)	ref	--		
COVID × Water pipe (*p* = 0.014)				
During-pandemic-hot-water pipe			3.98 (1.84, 8.61)	<0.001
During-pandemic-cold-water pipe			3.12 (1.38, 7.06)	0.002
Before-pandemic-hot-water pipe			2.91 (1.36, 6.23)	0.002
Before-pandemic-cold-water pipe			ref	--
During-pandemic-hot-water pipe			1.36 (0.17, 2.23)	0.365
During-pandemic-cold-water pipe			1.07 (0.61, 1.88)	0.990
Before-pandemic-hot-water pipe			ref	--
During-pandemic-hot-water pipe			1.27 (0.73, 2.22)	0.682
During-pandemic-cold-water pipe			ref	--

Notes: Sample aggregated by quarter resulting in 4897 observations. Due to missing water line temperature data, only 4399 observations (89.8%) were used in this analysis. ^1^ Scheffe test was used to adjust for multiple comparison.

**Table 4 ijerph-20-06885-t004:** Relative Risk(s) of Environmental Samples Testing Positive for *Legionella* (Non-SG1).

Effects	Risk Ratio (95% CI)	*p*-Value	Risk Ratio (95% adj.^1^ CI)	adj.^1^ *p*-Value
COVID (*p* < 0.001)				
Before COVID (3 January 2018–30 January 2020)	ref	--		
During COVID (1 February 2020–17 November 2020)	2.12 (1.74, 2.59)	<0.001		
Water pipe (*p* < 0.001)				
Hot-water pipe	1.59 (1.31, 1.93)	<0.001		
Cold-water pipe	ref	--		
Season (*p* = 0.373)				
Cold seasons (11/1–4/30)	1.17 (0.83, 1.65)	0.373		
Warm seasons (5/1–10/31)	ref	--		
COVID × Water pipe (*p* = 0.013)				
During-pandemic-hot-water pipe			3.38 (2.19, 5.21)	<0.001
During-pandemic-cold-water pipe			2.72 (1.72, 4.29)	<0.001
Before-pandemic-hot-water pipe			2.04 (1.33, 3.13)	<0.001
Before-pandemic-cold-water pipe			ref	--
During-pandemic-hot-water pipe			1.66 (1.20, 2.30)	<0.001
During-pandemic-cold-water pipe			1.33 (0.93, 1.91)	0.169
Before-pandemic-hot-water pipe			ref	--
During-pandemic-hot-water pipe			1.24 (0.87, 1.77)	0.391
During-pandemic-cold-water pipe			ref	--

Notes: Sample aggregated by quarter resulting in 5174 observations. Due to missing water line temperature data, only 4663 observations (90.1%) were used in this analysis. ^1^ Scheffe test was used to adjust for multiple comparison.

**Table 5 ijerph-20-06885-t005:** Disinfectant Measurements Within Control Limit During the COVID-19 Pandemic vs. Before ^1^ by Water System.

Water System ^2^	Measurement Type	Control Limit	Total Measurements		Average Measurements per Month ^3^		Measurements Below Control Limit	
			Before Pandemic	During Pandemic	Before Pandemic	During Pandemic	Before Pandemic	During Pandemic
Hot Water	Free Chlorine	<0.4 mg/L	17,539	6050	702	550	8269 (47%)	2908 (48%)
	Total Chlorine	<1.0 mg/L	17,393	6054	696	550	10,658 (61%)	3887 (64%)
Cold Water	Free Chlorine	<0.5 mg/L	17,047	5964	682	542	7211 (42%)	2508 (42%)
	Total Chlorine	<1.0 mg/L	16,923	5911	677	537	4814 (28%)	1599 (27%)

^1^ The during-pandemic period covers 11 months, while the before-pandemic period represents 25 months. ^2^ A total of 725 facilities measured disinfectant in their hot-water system and 760 facilities measured disinfectant in their cold-water system. ^3^ The column includes all facilities in the calculation that measured the disinfectant in the water system rather than average measurements per month per facility.

**Table 6 ijerph-20-06885-t006:** Temperatures Below Control Limit During COVID-19 Pandemic vs. Before ^1^ by Thermostatic Mixing Valve (TMV) Status and Sample Location.

Sample Location	Measurement Type	Lower Control Limit	Total Measurements		Average Measurements Per Month		Measurements Below Control Limit	
			Before Pandemic	During Pandemic	Before Pandemic	During Pandemic	Before Pandemic	During Pandemic
Supply	TMV	<124°F	9827	1926	393	175	344 (4%)	64 (3%)
	No TMV	<124°F	32,066	5189	1283	471	1389 (4%)	237 (5%)
Return	TMV	<118°F	10,099	1940	404	176	409 (4%)	64 (3%)
	No TMV	<118°F	32,247	5242	1290	477	2077 (6%)	231 (4%)

^1^ The during-pandemic period covers 11 months, while the before-pandemic period represents 25 months.

## Data Availability

Restrictions apply to the availability of these data. Data were obtained from a private entity and are available from the authors with the permission of the NLO and in accordance with data use agreement(s).

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
