# Peer review of "Effects of the COVID-19 Pandemic on Legionella Water Management Program Performance across a United States Lodging Organization"

_ijerph, 2023, doi:10.3390/ijerph20196885_

Round 1
Reviewer 1 Report
1. The manuscript mentions that developing and maintaining a WMP is a multi-step process that requires continuous review, but it does not elaborate on what these steps are or how often the review should take place. Providing more information on this aspect would enhance the understanding of the topic.
2. While it is mentioned that the COVID-19 pandemic had a dramatic impact on the built environment, including building water systems and devices, there is no further explanation or evidence provided to support this claim. Adding specific examples or studies that demonstrate the impact of the pandemic on water usage patterns would strengthen this statement.
3. The manuscript states that reduced water usage can make systems unsafe for use and create conditions favorable for Legionella growth such as stagnation or depleted disinfectant residual [13-15]. It would be beneficial to include specific studies or research findings that support this claim.
4. The lack of water quality data collected from outside of the national lodging organization's facilities is a significant limitation. This omission prevents a comprehensive understanding of the overall water quality and potential sources of contamination.
5. The use of the cold water test point as a proxy for source water is a potential confounder. While efforts were made to choose a location as close to the building's incoming water main as possible, there may still be variations in water quality between this test point and the actual source.
6. The potential influence of building-specific Legionella growth risk factors on the cold water test point should be considered. Factors such as reduced water usage during the pandemic could impact the results and introduce bias.
If the word "best" serves here to qualify results or methods, it will be considered hype and should be avoided. Consider replacing it with "optimal" or "reasonable" or just removing it.
Replace likely redundant "time period" with just "period".
You used some difficult words like "establish". Try using simple synonyms, like "set" because most readers of scientific papers are not native English speakers.
Avoid constructions with "there is" since they obscure the main subject and action of a sentence.
Author Response
- The manuscript mentions that developing and maintaining a WMP is a multi-step process that requires continuous review, but it does not elaborate on what these steps are or how often the review should take place. Providing more information on this aspect would enhance the understanding of the topic.
Thank you for your comment. We added language to the manuscript describing how often a water management program should be reviewed and cited CDC’s Legionella: Developing a Water Management Program | CDC.
- While it is mentioned that the COVID-19 pandemic had a dramatic impact on the built environment, including building water systems and devices, there is no further explanation or evidence provided to support this claim. Adding specific examples or studies that demonstrate the impact of the pandemic on water usage patterns would strengthen this statement.
Thank you for your comment. We added a reference in the manuscript citing available studies which are unfortunately limited due to the complexity of studying long-term stagnation.
Molina JJ, Bennassar M, Palacio E, Crespi S. Impact of prolonged hotel closures during the COVID-19 pandemic on Legionella infection risks. Front Microbiol. 2023 Feb 24;14:1136668. doi: 10.3389/fmicb.2023.1136668. PMID: 36910223; PMCID: PMC9998907.
- The manuscript states that reduced water usage can make systems unsafe for use and create conditions favorable for Legionella growth such as stagnation or depleted disinfectant residual [13-15]. It would be beneficial to include specific studies or research findings that support this claim.
Thank you for your comment. We added a reference that includes the latest studies in this area:
Vosloo S, Huo L, Chauhan U, Cotto I, Gincley B, Vilardi KJ, Yoon B, Bian K, Gabrielli M, Pieper KJ, Stubbins A, Pinto AJ. Gradual Recovery of Building Plumbing-Associated Microbial Communities after Extended Periods of Altered Water Demand during the COVID-19 Pandemic. Environ Sci Technol. 2023 Feb 28;57(8):3248-3259. doi: 10.1021/acs.est.2c07333. Epub 2023 Feb 16. PMID: 36795589; PMCID: PMC9969676.
- The lack of water quality data collected from outside of the national lodging organization's facilities is a significant limitation. This omission prevents a comprehensive understanding of the overall water quality and potential sources of contamination.
Thank you for your comment. We agree that including water quality data from outside the national lodging organization’s facilities would benefit the manuscript. Unfortunately, the authors were unable to link facility-level data with municipal water system data as the facilities were not identifiable and location information was not available. Cold water test points nearest the water main often serve as proxies for cold water entering a building. The authors feel our findings underscore the importance of monitoring incoming water quality by establishing a cold-water control point that includes water quality parameter monitoring and consideration of Legionella testing.
- The use of the cold water test point as a proxy for source water is a potential confounder. While efforts were made to choose a location as close to the building's incoming water main as possible, there may still be variations in water quality between this test point and the actual source.
Thank you for your comment. We concur and recognize this as a limitation to the study. This important limitation is detailed with the following language:
The findings in this study are subject to important limitations. The NLO did not have water quality data collected from outside of their facilities. To minimize this confounder, the cold water test point sample served as a proxy for source water and the location was chosen to be as proximal to the building incoming water main as possible. The cold water test point was potentially subject to building-specific Legionella growth risk factors such as reduced water usage, which was commonly experienced during the pandemic.
- The potential influence of building-specific Legionella growth risk factors on the cold water test point should be considered. Factors such as reduced water usage during the pandemic could impact the results and introduce bias.
Thank you for your comment. We concur and recognize this as a limitation to the study. This important limitation is detailed with the following language:
The findings in this study are subject to important limitations. The NLO did not have water quality data collected from outside of their facilities. To minimize this confounder, the cold water test point sample served as a proxy for source water and the location was chosen to be as proximal to the building incoming water main as possible. The cold water test point was potentially subject to building specific Legionella growth risk factors such as reduced water usage, which was commonly experienced during the pandemic.
Comments on the Quality of English Language
1. If the word "best" serves here to qualify results or methods, it will be considered hype and should be avoided. Consider replacing it with "optimal" or "reasonable" or just removing it.
Thank you for your comment. We accepted this comment and made edits (changed “best” to “optimal” within the manuscript.
2. Replace likely redundant "time period" with just "period".
Thank you for your comment. We accepted this comment and made edits within the manuscript.
3. You used some difficult words like "establish". Try using simple synonyms, like "set" because most readers of scientific papers are not native English speakers.
Thank you for your comment. We appreciate your recommendation to improve readability for non-English speakers. We changed “establish” to “set” or another synonym in all instances where doing so does not change the meaning.
4. Avoid constructions with "there is" since they obscure the main subject and action of a sentence.
Thank you for your comment. We accepted this comment and removed “there is” from the manuscript.
Reviewer 2 Report
The authors present an interesting paper on Legionella positivity in buildings before and during the COVID pandemic. A strength of the paper is the large data set used. The paper is well written and presented.
I have a few comments that require addressing before the paper can be considered for publication:
1. A major limitation not discussed in the paper is the reliance on positivity data for Legionella culture alone. Why have the authors not considered the amount of Legionella detected in addition to the positivity rate? The limits of detection of the ELITE lab methods used have not been described so is a positive test 1 CFU per litre of water or 1000? Please at the minimum describe the test limitations and consider this as part of the overall limitations of the study.
2. The Findings around cold water being just as likely to test positive for legionella than hot water is interesting. Can the authors please expand on why this may be the case given cold water systems are typically considered much lower risk for Legionella growth?
3. Line 39: The statement ‘However, all Legionella can cause disease’ is incorrect. There is insufficient evidence to say that the 58+ species of Legionella can all cause disease. I believe only 25 are actually linked to disease. Please correct or provide a reference which refutes my statement.
4. Line 40: The statement ‘more than 95% of persons who contract LD are hospitalised’ – is this figure accurate? If so, is it based on US data only, if so please state. To be diagnosed with LD the chances the patient is already in hospital for respiratory illness (or in hospital for another ailment and caught nosocomial LD). How many cases are diagnosed in the community but not hospitalised? Seems a bit of an irrelevant statistic if you follow me?
5. Information of the flushing regime should be included in the methods, not the discussion
Author Response
I have a few comments that require addressing before the paper can be considered for publication:
- A major limitation not discussed in the paper is the reliance on positivity data for Legionella culture alone. Why have the authors not considered the amount of Legionella detected in addition to the positivity rate? The limits of detection of the ELITE lab methods used have not been described so is a positive test 1 CFU per litre of water or 1000? Please at the minimum describe the test limitations and consider this as part of the overall limitations of the study.
Thank you for your comment. Although we do know that all water samples were tested by an ELITE Program member laboratory using traditional spread-plate culture, there is likely to have been variation in the sample collection methods (e.g., volume collected) and testing methods (e.g., volume of sample processed, direct vs. concentrate plating, plate treatment) that can impact the LOD on an individual sample basis. Unfortunately, the LOD for each sample is not available. This limitation has been added to the paper discussion.
Rather than analyzing Legionella concentration, a logarithmic approach was undertaken. Previous studies have observed greater consistency among ELITE Program member laboratories for qualitative results (e.g., detection of Legionella) than for qualitative results. This consideration and the appropriate citation have been added to the methods.
Lucas CE, Taylor TH Jr, Fields BS. Accuracy and precision of Legionella isolation by US laboratories in the ELITE program pilot study. Water Res. 2011 Oct 1;45(15):4428-36. doi: 10.1016/j.watres.2011.05.030. Epub 2011 Jun 7. PMID: 21726887.
- The Findings around cold water being just as likely to test positive for legionella than hot water is interesting. Can the authors please expand on why this may be the case given cold water systems are typically considered much lower risk for Legionella growth?
Thank you for your comment. We currently have language in the paper that recognizes the potential role of altered water usage patterns within the public water distribution system (lines 336-338). We also discuss in the limitations that the cold-water test point (lines 387-392) was potentially subject to building specific Legionella growth risk factors such as reduced water usage, which was commonly experienced during the pandemic. Additionally, the limitations associated with the disinfectant values (394-405) did not allow for the type of analysis to determine whether disinfectant levels were noticeably different during the pandemic (low or no water use) than before the pandemic when water use was normal.
- Line 39: The statement ‘However, all Legionella can cause disease’ is incorrect. There is insufficient evidence to say that the 58+ species of Legionella can all cause disease. I believe only 25 are actually linked to disease. Please correct or provide a reference which refutes my statement.
Thank you for the comment. We have changed the text to reflect that while many non-pneumophila species can cause disease, saying all is likely not correct.
- Line 40: The statement ‘more than 95% of persons who contract LD are hospitalised’ – is this figure accurate? If so, is it based on US data only, if so please state. To be diagnosed with LD the chances the patient is already in hospital for respiratory illness (or in hospital for another ailment and caught nosocomial LD). How many cases are diagnosed in the community but not hospitalised? Seems a bit of an irrelevant statistic if you follow me?
Thank you for the comment. We have changed the text to indicate that this figure represents only reported cases in the United States. We have also included a citation supporting this statistic. Your point about diagnosed cases having a bias towards hospitalization is well taken.
- Information of the flushing regime should be included in the methods, not the discussion
Thank you for your comment. We’ve added additional text to describe the flushing protocol in the methods.
Reviewer 3 Report
The influence of reduced occupancy of buildings on water quality and Legionella risk was yet another knock-on effect of the pandemic. While this potentially was overlooked in some sectors it is a positive, that was highlighted by this paper, that the NLO who provided the data had the forethought and a clear WMP strategy. The information in this paper is valuable and worth sharing with others, in this sector and with other sectors, including that a simple-to-implement flushing regime can have positive impact in many cases.
I have a small number of questions or comments.
Minor point - I suggest that for a wider international audience that temperatures (lines 119-130) are presented in Celsius as well as Fahrenheit.
In the Methods section line 329 on you mention intervention when a positive sample found - are any details available of what this is? What was considered an action level (colony forming units per litre) and what done? I accept intervention will differ from site to site but presumably the NLO's generic WMP would be a source of examples and I believe would add context to the paper. Similarly, you present data as 'Legionella positive'. Can you say what the lower limits of detection (LLOD) are? Presumably 'Legionella positive' is anything above the LLOD. Control limits are given for chlorine and temperature so this additional information I believe is relevant. I accept that such data will be available in CDC and ASHRAE guidance but a short summary helps the reader.
Lines 367- 8 you make a good point about not making the assumption that risk is lower in cooler months.
Line 341 - any ideas (that could be put into the paper) as to why incoming cold water supply was 3x more likely to be Legionella positive during the pandemic? Presumably no difference in treatment regime, so could it be to do with lower volumes going in and effects of stagnation or build up of biofilm in supply pipes?
Author Response
- Minor point - I suggest that for a wider international audience that temperatures (lines 119-130) are presented in Celsius as well as Fahrenheit.
Thank you for your comment. We made the recommended changes and included Celsius.
- In the Methods section line 329 on you mention intervention when a positive sample found - are any details available of what this is? What was considered an action level (colony forming units per litre) and what done? I accept intervention will differ from site to site but presumably the NLO's generic WMP would be a source of examples and I believe would add context to the paper. Similarly, you present data as 'Legionella positive'. Can you say what the lower limits of detection (LLOD) are? Presumably 'Legionella positive' is anything above the LLOD. Control limits are given for chlorine and temperature so this additional information I believe is relevant. I accept that such data will be available in CDC and ASHRAE guidance but a short summary helps the reader.
Thank you for your comments. NLO facilities respond to Legionella detections in a tiered approach according to concentration values if any Legionella is detected. Response activities could include confirming that equipment is in good working order, reviewing records to confirm that the WMP was implemented, adjusting the control activities (e.g., increasing flushing frequency), implementation of remedial treatment, or other activities to address Legionella and improve the WMP performance. While precise response activities are not available to be shared, language about the tiered approach to response according to concentration has been added.
Unfortunately, the LOD likely varied by sample and is not available. This clarification has been added to the methods and discussed as a limitation.
- Lines 367- 8 you make a good point about not making the assumption that risk is lower in cooler months
Thank you for your comment.
- Line 341 - any ideas (that could be put into the paper) as to why incoming cold water supply was 3x more likely to be Legionella positive during the pandemic? Presumably no difference in treatment regime, so could it be to do with lower volumes going in and effects of stagnation or build up of biofilm in supply pipes?
Thank you for your comment. We currently have language in the paper that recognizes the potential role of altered water usage patterns within the public water distribution system (lines 336-338). We also discuss in the limitations that the cold-water test point (lines 387-392) was potentially subject to building specific Legionella growth risk factors such as reduced water usage, which was commonly experienced during the pandemic. Additionally, the limitations associated with the disinfectant values (394-405) did not allow for the type of analysis to determine whether disinfectant levels were noticeably different during the pandemic (low or no water use) than before the pandemic when water use was normal.
Reviewer 4 Report
A very well written article. A very important problem with Legionella bacteria in water distribution systems was presented concisely and logically. It just needs to be edited.
Author Response
- A very well written article. A very important problem with Legionella bacteria in water distribution systems was presented concisely and logically. It just needs to be edited.
Thank you for your comment. We worked to improve the readability of the paper making several edits. We also made grammar edits to improve the understanding for non-English speakers based on reviewer feedback.
Round 2
Reviewer 1 Report
The methodology of the manuscript should be adapted based on the following:
Literature Review:
1 COVID-19 pandemic's impact on building water quality
2 Legionella growth and associated risks
2 Flushing as a control measure for Legionella
Study Design:
1 Selection of buildings and sampling sites
2 Data collection methods
3 Water sampling protocols
4 Environmental parameters monitoring
5 Occupancy data collection (if applicable
6 Sample size determination
7 Data analysis plan
8 Evaluation of Building Water Quality:
9 Microbiological analysis
10 Enumeration of Legionella species
Identification of other potential pathogens
Physicochemical analysis
Temperature, pH, and disinfectant levels monitoring
Nutrient levels assessment (if applicable)
Comparison with pre-pandemic data (if available)
Minor changes are required.
Author Response
Thank you for your comments. We are having trouble understanding this round of comments and the associated recommendations. We have responded based on our interpretation of the comments received.
Literature review:
1 COVID-19 pandemic's impact on building water quality
2 Legionella growth and associated risks
3 Flushing as a control measure for Legionella
- We added a reference in the manuscript during the previous round of revisions citing the pandemic’s impact on building water quality. Studies are unfortunately limited due to the complexity of studying long-term stagnation.
Molina JJ, Bennassar M, Palacio E, Crespi S. Impact of prolonged hotel closures during the COVID-19 pandemic on Legionella infection risks. Front Microbiol. 2023 Feb 24;14:1136668. doi: 10.3389/fmicb.2023.1136668. PMID: 36910223; PMCID: PMC9998907
- The manuscript includes references that site Legionella growth and associated risks (Line 485).
- The manuscript includes mention of available guidance that includes flushing as a control measure for Legionella (Line 71) & (Line 304).
- Recommended steps for returning to normal occupancy or reopening after a period of no or low water usage include using or creating a WMP, flushing, and ensuring all potable and non-potable devices are properly maintained according to manufacturer recommendations [16,18,19,20].
- Routine flushing is a recommended practice as part of a WMP to maintain water quality parameters within control limits during periods of no or low water usage. However, a lack of real-world evidence exists on the use of flushing as an effective control measure to reduce the risk of Legionella growth during long-term periods of unusually low water usage [11,20]
Study Design:
1 Selection of buildings and sampling sites
2 Data collection methods
3 Water sampling protocols
4 Environmental parameters monitoring
5 Occupancy data collection (if applicable
6 Sample size determination
7 Data analysis plan
8 Evaluation of Building Water Quality:
9 Microbiological analysis
10 Enumeration of Legionella species
Identification of other potential pathogens
Physicochemical analysis
Temperature, pH, and disinfectant levels monitoring
Nutrient levels assessment (if applicable)
Comparison with pre-pandemic data (if available)
1. Thank you for your comment regarding study design. We’ve included here further clarification about our attempts to address these study design considerations. It is important to note that we were provided access to data as collected during ongoing water management program activities across many facilities and received by the facilities’ national organization (Line 81). The authors did not have the ability to direct methodology employed by the national lodging organization nor it’s member facilities. We describe in the methods what is known about NLO data collection methods, water sampling protocols, and environmental parameters monitoring. We describe within the limitations that occupancy data are unavailable, what is known about enumeration of Legionella species, constraints associated with temperature, and disinfectant data. Other water parameter data, such as pH and nutrient levels, were not available for this analysis. The inability to compare pre-pandemic and pandemic temperature and disinfectant levels is also addressed within the limitations (Line 414).